# Road Surface Anomaly Assessment Using Low-Cost Accelerometers: A Machine Learning Approach

**DOI:** 10.3390/s22103788

**Published:** 2022-05-16

**Authors:** Alessio Martinelli, Monica Meocci, Marco Dolfi, Valentina Branzi, Simone Morosi, Fabrizio Argenti, Lorenzo Berzi, Tommaso Consumi

**Affiliations:** 1Telespazio, Via Tiburtina 965, 00156 Rome, Italy; alessio.martinelli@telespazio.com; 2Department of Civil and Environmental Engineering, University of Florence, 50139 Florence, Italy; monica.meocci@unifi.it (M.M.); valentina.branzi@unifi.it (V.B.); 3Department of Information Engineering, University of Florence, 50139 Florence, Italy; marco.dolfi@unifi.it (M.D.); fabrizio.argenti@unifi.it (F.A.); tommaso.consumi@unifi.it (T.C.); 4Department of Industrial Engineering, University of Florence, 50139 Florence, Italy; lorenzo.berzi@unifi.it

**Keywords:** low-cost road surface monitoring, accelerometer, digital signal processing, machine learning, pavement distress classification

## Abstract

Roads are a strategic asset of a country and are of great importance for the movement of passengers and goods. Increasing traffic volume and load, together with the aging of roads, creates various types of anomalies on the road surface. This work proposes a low-cost system for real-time screening of road pavement conditions. Acceleration signals provided by on-car sensors are processed in the time–frequency domain in order to extract information about the condition of the road surface. More specifically, a short-time Fourier transform is used, and significant features, such as the coefficient of variation and the entropy computed over the energy of segments of the signal, are exploited to distinguish between well-localized pavement distresses caused by potholes and manhole covers and spread distress due to fatigue cracking and rutting. The extracted features are fed to supervised machine learning classifiers in order to distinguish the pavement distresses. System performance is assessed using real data, collected by sensors located on the car’s dashboard and floorboard and manually labeled. The experimental results show that the proposed system is effective at detecting the presence and the type of distress with high classification rates.

## 1. Introduction

Since ancient times, roads have always represented the fulcrum of the economic development of a country. Roads represented the easiest and fastest way to extend military and economic power across territories. Nowadays, roads have not lost their strategic importance and still represent the main “way” to provide mobility and connectivity over short and medium distances. These aspects induced the World Bank to select the density of paved roads in good condition as an indicator of the “economic strength” of a country [1] and, hence, its competitiveness [2].

The extension of the road network, the increase in volume and traffic load, effects of climate change [3] and road age are all factors that create different types of anomalies on the road surface (such as potholes, cracks, rutting, etc.); this fact is true for all types of roads (urban, suburban and highway) and is common all over the world.

Bad road surfaces are one of the main concerns of public administrations regarding the safety of road users. As an example, a survey on road safety published by the Regional Road Safety Monitoring Centre (CMRSS) of the Tuscany region (Italy) highlighted that the presence of potholes and other road pavement distresses are perceived as the most dangerous elements for 41.6% of mopeds and/or motorcycle drivers and 32.8% of the drivers of four-wheeled vehicles (car, minicar, van, truck and coach) [4]. Dangerous road surfaces could be the cause of unsafe and uncomfortable transportation as well as expensive lawsuits and damage claims [5]. In these terms, there are accidents caused directly by the poor condition of the road, but there are also accidents caused by drivers’ behavior in reaction to the poor condition of the road. Across Europe, this accounts for about 4% of accidents. Different percentages characterize different countries whose roads are characterized by various levels of maintenance. In the USA, it is estimated that one-third of traffic fatalities occur due to poor road conditions [6]; this results in over 10,000 fatalities per year. Across the world, the data concerning accidents that occurred due to poor road surface condition ranges from low percentages (high-income countries) to very high values characterizing low- and medium-income countries (e.g., in India more than 50% of accidents occur due to the presence of potholes). According to Italian data from 2019, 1349 accidents including fatalities and serious injuries occurred due to road surface damage [7]. No official data are available to estimate the number of lawsuits per year due to small injuries or crashes resulting in property damage only. Therefore, maintaining an acceptable level of efficiency for the entire road network by adopting effective road pavement monitoring and maintenance programs are some of the main challenges for Road Authorities (RAs) across the world [5]. Keeping road surfaces in good condition requires RAs to invest significant budgets and resources on decision-support systems. For instance, it is necessary to determine which roads need to be restored and repaired first by evaluating both economic costs and road safety criteria.

The objective of this work is to develop a low-cost and reliable system able to carry out real-time screening of road pavement conditions. The data are acquired with accelerometers mounted on a vehicle, and the data are analyzed by machine learning.

Traditional methods basically rely on human observation, such as Pavement Condition Index (PCI) surveys, which are an international reference for all RAs and technicians [8]. However, the process is prone to subjective evaluation and is time-consuming and risky for operators on the road. Newer tools and high-performance methodologies use videos and image analysis to identify and classify distresses on the pavement surface with more reliability; however, they currently involve high implementation costs. For example, automated technologies such as Ground Penetrating Radar, Laser Road Imaging systems and high-performance sensors coupled with high-resolution cameras are the most-used monitoring processes today. However, the high costs associated with such technologies are an important limit for road authorities, especially those of small municipalities that may lack the budget or professional skills to carry out regular monitoring. Vibration-based approaches could represent a suitable trade-off to solve the high-cost monitoring process based on video/image processing and the RAs’ need to routinely monitor the road network. Inertial sensor (e.g., accelerometers)-based methods use vehicle vibrations to detect road surface anomalies. Albeit simpler, such methods do not have the same implementation status as image processing systems, which are more complete and already available in commercial products. The literature also demonstrates the relevance of methods based on tracking the real-time evolution of dynamic parameters of structures in order to extract damage-sensitive features (DSFs) through the analysis of natural frequencies and modal shapes constituting the “eigenspace” of the system [9]. In this context, real-time damage identification can be carried out by means of eigen-perturbation techniques, which are computationally more efficient than other approaches such as Kalman filtering, which becomes resource intensive in cases of many degrees of freedom [10]. According to the literature, such techniques can be adapted to detect road surface irregularities, but power spectral density (PSD) is still adopted as a convenient way to identify road status and roughness [11]. To fill the gap in the literature findings, it is important to provide RAs with tools able to reliably screen the road pavement conditions of the entire road network while being user-friendly, high performance and low cost [12].

The literature review also showed the main limit of vibration-based approaches as the inability to extract (from the measured data) information about the type of distress on the road surface. Additionally, it highlighted that this method cannot measure road-surface damage in areas other than the vehicle wheel paths, with the consequence of not being able to identify the size of the road-surface damage across the entire road section [13].

To address these issues, this paper presents an approach based on processing the data recorded by accelerometers mounted on a car that routinely passes over the road network. The proposed techniques, applied in a mobile crowdsensing framework (e.g., data recorded from taxi, public officials, public transportation, etc.) could play an important role in obtaining a large quantity of data concerning vehicle vibration. For example, a mobile phone app (using phones’ accelerometers) could “describe the road network maintenance condition” similar to Google Maps for traffic. Recent studies used this type of data sourcing to monitor different aspects concerning roads, traffic, traffic noise, etc. [14]. Differently from the literature findings of vibration-based methodologies, the proposed method aims both to detect the presence of distresses on the road surface and to identify the distress type and classify its severity. This is all conducted based on the signals’ energy content. To achieve this purpose, acceleration signals provided by on-car sensors are analyzed through short-time Fourier transform to highlight the energy of the signal in the time–frequency domain. Then, significant features such as the coefficient of variation and the entropy are extracted from the signal energy and exploited to distinguish between different pavement distresses. Supervised learning algorithms are used to classify three classes of pavement conditions: the first one refers to well-localized (in space) distresses caused by potholes and manhole covers, the second one relates to spread distresses due to fatigue cracking and rutting, and the third one indicates the absence of distresses. A measurement campaign was performed to collect real data and to test the performance of the classification algorithms by comparing two sensor locations: the dashboard and the floorboard of the car.

The paper is organized as follows. Section 2 provides a review of the state-of-the-art concerning automated or semi-automated road distress detection and classification. Section 3 describes the quality of the road surface by characterizing prospective pavement distresses. Section 4 details the car model and its components involved in the system assessment. Section 5 describes the system proposed for the evaluation of road surface quality, focusing in particular on the subsystems of signal processing, feature extraction and pavement distress classification. Section 6 provides the experimental setup to test the proposed system based on real data, whereas Section 7 shows and discusses the experimental results relative to the time–frequency analysis of the acceleration signal and the classification of pavement distresses. Finally, Section 8 summarizes the conclusions of this work.

## 2. Related Works

There are several protocols for road distress pavement detection. Road pavement surveys can be conducted with low- or high-performance methodologies. The former, generally conducted by visual inspection (such as PCI surveys [8]), are laborious, time-consuming and prone to the subjectivity of the inspectors. These methodologies are very similar, and they also expose the inspectors to dangerous working conditions. On the other hand, high-performance procedures instead offer automated or semi-automated detection solutions, which minimize subjectivity and improve productivity. Vision-based methodologies have had a great impact on the detection and classification of road anomalies (especially when coupled with high-performance sensors). They allow capturing all distresses within the road section with their exact location and severity. Despite their accuracy, the high cost of the equipment represents the main disadvantage to large-scale implementation on the road network [15,16].

In recent times, many studies have focused on detecting road pavement distresses, using different technological solutions both for data collection and evaluation in order to speed up maintenance and repair. More and more efforts have been made to implement more advanced and effective low-cost monitoring systems to help RAs routinely determine pavement conditions along the road network. Tools and procedures based on artificial vision, shape segmentation and evaluation of pavement texture have been implemented, as well as more recent and elaborate methods based on stereo sensors and artificial vision and image processing algorithms such as 3D scans [16,17,18,19,20,21]. The cost of these processes prevent carrying out widespread surveys on a short time basis because the consequent costs are not sustainable by the RAs. Therefore, although effective, their use is limited along minor roads (e.g., urban/local rural roads), which represent the most common type of road [22]. Many researchers have exploited acceleration signals for motion detection and classification in pedestrian contexts [23,24,25], whereas few studies have analyzed the results obtained from several vehicles equipped with accelerometers for data acquisition and specific algorithms for data processing. From this perspective, in recent years the reliability of accelerometers, GPS and even smartphone sensors has been exploited to address this issue. Mednis et al. [26] evaluated pothole detection on real data acquired using different Android smartphones with great consistency. Mohan et al. [27] monitored the roads and traffic conditions using mobile smartphones equipped with sensors such as GPS, accelerometers and microphones, focusing on sensors capable of detecting bumps, potholes, braking and honking. Pothole Patrol System [5] was instead based on a simple machine learning approach to filter accelerations and velocities in order to reduce the number of false positives in pothole detection. The study was conducted by installing sensors inside the taxis circulating downtown Boston. In the last two studies, threshold-based heuristics were proposed to classify road anomalies, while in [5], the authors used signals obtained from triaxial accelerometers and GPS (with sampling rates of 380 Hz and 1 Hz, respectively) installed in a fixed position in seven taxis; in [27], the authors introduced the use of smartphones to monitor the city’s traffic conditions in addition to those of road pavements. Recently, Meocci et al. [12,28,29] explored the crowdsourcing and “participatory sensing” approach where data recorded by black boxes were collected from all vehicles on the road network, shared and added as a layer to existing navigation systems, such as Waze, Google Maps, etc. The proposed system used real-time traffic information in order to localize road pavement distress across the road network: when a vehicle passes over a road surface anomaly (e.g., a pothole or a bump), the accelerometers mounted on the vehicle record the event by linking it to position information provided by the GPS receiver. This research also defined the thresholds of severity levels with reference to the PCI classification process [12]. Although the detection algorithm allows understanding some information from the recorded signal (e.g., length of the distress as a function of GPS information and distance between two axes passage), it does not allow the distresses on the road pavement surface to be classified. Conversely, vision-based tools are able to process image texture, dimensions and shape. Due to these limitations, the development of these low-cost methods has not spread over time even though the first studies are now dated [30,31,32]. In this context, the study conducted by Lekshmipathy (2020) concludes by stating that vibration-based analysis is sufficient only for routine monitoring [33] but needs to be coupled with more advanced technologies when RAs define how to repair the distresses previously detected. Indeed, the type, extent and severity of road pavement anomalies are crucial characteristics for assessing the actual condition of road sections, as well as to determine the most appropriate maintenance solutions. From a cost-effectiveness point of view, several gaps were found in the detection of fatigue cracking anomalies by means of low-cost and user-friendly methodologies. The literature proposes numerous suggestions for the detection of these types of distress, including cameras, 3D sensors, microphones, sonar, laser, pressure sensors and accelerometer, with the first two technologies also commonly found in commercial solutions.

The literature review [15] highlighted the importance of automated road pavement inspection methods being dependent on the type of road distress considered. For example, cracks are usually detected and classified through imaging techniques or ground penetrating radar and infrared. Potholes or patches are detected and classified with vision-based techniques, laser/sonar, or even by vibrations due to the vertical drop of the surface. Surface deformations are instead detected and classified with 3D surface analysis similar to rutting detection. Multiple distresses on the road surface are usually detected and classified with 3D vision-based or by jointly using different techniques.

To summarize this literature review, road surface distress detection using accelerometers is a technique that needs to be further studied and improved [34]. It is indeed necessary to offer to RAs a reliable and low-cost process able to screen the entire road network in a short time and to detect and classify the different distresses on the road surface in terms of severity and type. Most of the related studies focus on binary classification of inertial signals and do not implement multi-class classification for different types of road surface degradation. In addition, the data collected from the acceleration signal in the direction of gravity consider only threshold-based heuristics focusing on mean and standard deviation while ignoring higher order statistics. Therefore, the proposed approach is characterized by an inexpensive though reliable way for an accelerometer setup assessment to extract novel features to distinguish road pavement conditions, and the use of supervised machine learning for multi-class classification. In particular, it offers new insights in the set of features for developing the distress classification model that can lead to effective practices. Mastery of this low-cost and easy-to-use classification scheme will enable RAs to understand how to effectively implement policies that will positively impact their organizations and practitioners.

## 3. Road Surface Characterization

Well-maintained roads are mandatory to guarantee transport efficiency, adequate ride quality and road safety for all road users. Road pavement distress represents a natural process due to the effects of different factors, such as aging, traffic loads, adequate material properties and weather conditions. In 1970, the Highway Research Board defined surface distress as “any indication of poor or unfavorable pavement performance or signs of impending failure; any unsatisfactory performance of a pavement short of failure” [35]. Road pavement distresses can be classified according to different methodologies proposed by researchers and administrations. One of the most-used road surface assessments is the visual survey proposed by ASTM [8], in which the distress type, the severity and the extension are quantified (see also [36]). Based both on the assessment proposed by the ASTM and on distresses that can be detected by an inertial system, the proposed procedure considers only the following types of distress:potholes: small, bowl-shaped depressions in the pavement surface (usually less than 750 mm in diameter);fatigue cracking: interconnecting cracks caused by fatigue failure of the asphalt concrete surface under repeated traffic loading;others: any other anomalies that could generate a high level of roughness on the road surface, such as patches, manhole covers or more distresses together.

Many studies have analyzed in detail the mechanical models that explain how each distress is generated based on structural pavement responses and their severity. Although these studies detail the behavior of road pavement due to load conditions, they do not help with carrying out rapid and efficient monitoring.

## 4. The Car Model

In this study, an accelerometer is positioned within the car either on the dashboard or the floorboard (see Section 6). The exact location of the sensors within the vehicle yields a preliminary assessment of the transfer function between road and mounting point. In fact, all the elements placed between the origin of the solicitation (i.e., the contact point between tire and road surface) and the point of acquisition act as a filter that can potentially modify amplitude and phase of the transmitted signal. Due to the need for vehicle stability and comfort, suspension components are selected in order to absorb the energy of the solicitations by using elements with relevant damping properties. Analyzing the system from the point of contact with the road to the car body, the following components are involved with system vibrations:tire: can be represented by a spring and damper unit which connects the road contact point with the unsprung mass of the wheel subsystem. This system consists of the tire sidewall and is estimated according to rubber and tire ply characteristics;unsprung mass: includes masses unsprung relative to the car body but sprung in relation to the road contact point: elements such as the wheel rim, brake components, suspension elements and others;shock absorber: represented by elastic and damper elements, which, depending on mounting characteristics, can have linear or non-linear characteristics;car body: suspended on four different unsprung masses.

Due to such a complex structure, the use of multibody simulation models is usually required in order to consider all the degrees of freedom and all the suspension characteristics relevant to evaluate, through simulation, the quality of vehicle handling, stability and comfort [37]. Nevertheless, simplified models including a part of total vehicle mass and one single unsprung mass and its elements are still used for generic performance assessment, system dimensioning and evaluation. Such systems are usually called “quarter car models”. Road unevenness acts as a solicitation (displacement) on the vertical axis depending on vehicle position.

Assuming no loss of contact between road and tire, the model is able to evaluate the solicitations and displacements of mass depending on the magnitude and frequency of the vertical component. Such a system acts as a filter on road unevenness, and its characteristics (e.g., damping, phase and band-pass) influence the data acquired through on-board devices such as accelerometers. If the device is mounted on a component having its own degree of freedom in comparison with the vehicle chassis (e.g., a seat), a further sprung mass should be considered.

For the present application, it is assumed that the device is mounted on a rigid part of the vehicle chassis. The typical displacements accepted for road vehicles are usually frequency-dependent, and their amplitude is significantly reduced for frequencies above 4 Hz. The vehicle used for the case study was an M1, middle-class passenger vehicle (Suzuki Baleno). The vehicle was in perfect maintenance condition, and it was assumed that the suspension was fully efficient.

## 5. Classification of Road Pavement Distresses

In this Section, the proposed system for recognizing road pavement distresses is presented, and the relevant subsystems are detailed.

As shown in Figure 1, the overall system, from the acquisition of the acceleration signal to the classification of the road pavement distress, is composed of three subsystems—referred to as time–frequency analysis, feature extraction and distress classification—and detailed in the following.

The main objective of the proposed approach is to recognize the different pavement surface conditions from the acquired data. Thus, signal acquisition and processing is the first important task of the entire system. The signal provided by a 3-axial accelerometer is acquired at a given rate. The acceleration sensor can be rigidly attached to different placements within the car. Depending on sensor location, the signal provided by the accelerometer will have different patterns even over the same route.

Figure 2 shows some examples of acceleration signals based on different on-car sensor locations and associated with various road pavement conditions, where the z-component reveals acceleration in the direction perpendicular to the road surface and the x- and y-components lie parallel to the road surface.

As already observed, the acceleration signal can be modeled as a non-stationary process. For this reason, processing works on segments (along the time axis) of the signal. The presence of either repetitive or impulsive patterns in the acceleration signal has been characterized in the joint time–frequency domain, and a short-time Fourier transform (STFT) of the z-component has been used to identify the energy content at different frequencies along time.

In Figure 3, an example of an acceleration frame and its STFT are depicted. The STFT segments the acceleration frame in a given number of windows and performs the discrete Fourier transform (DFT) of the signal within each window. Its output can be represented as a matrix in which each column represents the DFT of a signal segment and is associated with a specific time. Windowing by means of tapered functions as well as overlapping segments can be used.

For our classification purposes, the STFT matrix is divided into submatrices, as shown in Figure 3b.

Let ${\mathrm{STFT}}_{p,u}\left(a_{z}\right)$ denote the short-time Fourier transform of the z-axis acceleration, where *p* and *u* are the indexes along the time and frequency axes, respectively, with 0≤p≤P−1 and 0≤u≤U−1, where *P* is the number of segments within the acquired frame and *U* is the number of frequency samples within each segment. Consider a set of adjacent STFT blocks having dimensions $P_{1}\times U_{1}$ and indexed by *i* and *j*, where 0≤i≤M−1, 0≤j≤N−1, with $M=P/P_{1}$ and $N=U/U_{1}$ the number of blocks along the time and frequency axes.

The energy contained in each submatrix (i,j) is computed as
(1)$$\begin{array}{cc} E_{i,j}=\sum_{p=iP_{1}}^{iP_{1}+P_{1}-1}\sum_{u=jU_{1}}^{jU_{1}+U_{1}-1} & |{\mathrm{STFT}}_{p,u}\left(a_{z}\right){|}^{2} \end{array}$$
The quantities in (Equation 1) are the input of the feature extraction subsystem, which aims to identify representative elements from the acceleration signal to classify different road pavement conditions.

The concept of sparsity is at the basis of the feature selection procedure. Isolated potholes are assumed to produce impulse-like acceleration signals that, in the time–frequency domain, are characterized by an energy blob widely spread in the frequency domain, but well-localized in the time domain. On the other hand, extensive rutting or pavement cracking are characterized by a cyclic roughness of the road surface, and thus, they are expected to produce a periodic-like signal with the duration spread over a long time interval but well-localized in the frequency domain.

In order to measure the dispersion of a set of discrete data *X*, the following two features have been considered in this study:
*coefficient of variation (CV)*: a dimensionless quantity defined by
(2)$$CV\left(X\right)=\frac{\sigma }{\mu },$$
where σ and μ are the standard deviation and the mean of the data contained in *X*;*entropy (H)*: assuming that *X* is composed of nonnegative data $\left\{x_{1},x_{2},\ldots ,x_{S}\right\}$ such that $\sum_{i=1}^{S}$$x_{i}=1$, its entropy is defined by
(3)$$H\left(X\right)=-\sum_{i=1}^{S}x_{i}\;log_{2}x_{i}.$$
It is well known from information theory, where entropy is used to measure the average information generated by a source, that entropy assumes its maximum when the data are uniform, whereas it tends to zero when most of the samples are zero and only a few of them dominate.

In order to identify sparse segments at a given frequency, the *CV* and *H* features can be computed along the rows (or the columns) of the blocks of the STFT. Consider the *i*-th row of blocks of the STFT matrix, with 0≤i≤M−1, and let us define
(4)$$CV_{i}=\frac{\sigma_{i}}{\mu_{i}},$$
where
(5)$$\begin{array}{cc} \mu_{i} & =\frac{1}{N}\sum_{j=0}^{N-1}E_{i,j},\\ \sigma_{i}^{2} & =\frac{1}{N}\sum_{j=0}^{N-1}{\left(E_{i,j}-\mu_{i}\right)}^{2}, \end{array}$$
and
(6)$$H_{i}=-\sum_{j=0}^{N-1}{\tilde{E}}_{i,j}\;log_{2}{\tilde{E}}_{i,j},$$
where ${\tilde{E}}_{i,j}$ is a scaled (normalized) version of $E_{i,j}$ such that $\sum_{j=0}^{N-1}$${\tilde{E}}_{i,j}=1$ for 0≤i≤M−1.

As already mentioned, CV and *H* are introduced to measure the flatness of a signal; by regarding the accelerometer as our data source, the entropy *H* (the coefficient of variation, CV) will be higher (lower) when the unpredictability of the generated signal is higher. For example, high entropy can be expected when the car is traveling along a road with good pavement, as shown in Figure 2a,b, where the accelerometer over undamaged pavement can be assimilated to a uniform noise source; on the other hand, when the road surface is damaged, the acceleration presents peaks, as shown in Figure 2c,d, which result in lower entropy.

Single values for the coefficient of variation and of entropy, simply referred to as CV and *H* in the following, can be associated to the entire STFT matrix relative to a frame of acceleration data. They are determined as follows:(7)$$CV=\sum_{i=0}^{M-1}\omega_{i}\;CV_{i},$$
(8)$$H=\sum_{i=0}^{M-1}\omega_{i}\;H_{i},$$
where the weights $\omega_{i}$ are computed as
(9)$$\omega_{i}=\frac{\sum_{j=0}^{N-1}E_{i,j}}{\sum_{i=0}^{M-1}\sum_{j=0}^{N-1}E_{i,j}},\;i=1,2,\ldots ,M,$$
and are introduced to give more emphasis to the rows of blocks in the STFT matrix containing more energy.

The CV and *H* features computed on frames of the acceleration signal are subsequently given as input to the distress classification subsystem. In this study, the following road pavement condition classes have been considered for the purpose of automatic classification:the short-time distress class: characterized by the presence of potholes or manhole covers; they are usually characterized by sharp edges and can generate high levels of depression/roughness in a small time interval relative to the duration of the observation window;the long-time distress class: characterized by road sections with fatigue cracking and rutting, typically caused by fatigue failure of the asphalt concrete surface under traffic loading. After repeated loading, the longitudinal cracks connect, forming many-sided, sharp-angled pieces that develop into a repetitive pattern. This anomaly creates oscillations in the acquired acceleration signal that are usually less intense but much longer lasting over time compared to the previous class;the no-distress class: refers to road sections in good condition without any visible or detectable road surface distress.

The main objective of the distress classification subsystem is to build up a mathematical model able to effectively map the features previously defined, i.e., CV and *H*, to the classes of road pavement conditions identified above. This goal is carried out by using supervised machine learning. For our study, three families of classification algorithms were chosen: Decision Tree (DT), Support Vector Machine (SVM) and *k*-Nearest Neighbor (*k*NN). They are popular and reliable techniques that have different characteristics in terms of memory usage, fitting and prediction speed, and predictive accuracy [38]. The DT algorithm generates a classification tree by training examples and distinguishes classes through splitting the tree’s path. It has fast prediction, low memory usage and fair predictive accuracy. The SVM algorithm seeks the hyperplane that maximizes the separation margin between the training data points of any class. Depending on the kernel function, the SVM can effectively separate the training data; it guarantees very good predictive accuracy, low memory usage and fast predictive speed, but only for a few support vectors, i.e., for a few features. The *k*NN algorithm tries to classify new data based on their distance from neighbors in the training set. It requires high memory usage, but provides high predictive accuracy and fast prediction speed with low-dimensional feature space.

## 6. Experimental Setup

This section describes the experimental setup used to test the proposed system and to assess its performance. The hardware components selected for experimentation are listed in the following. The acquisition system is driven by an *Arduino Yun* platform, based on the ATmega32u4 microcontroller and the Atheros AR9331 communication System-on-a-Chip (SoC), which has been chosen for its characteristics of low cost, versatility and ease of use when communicating with a large variety of sensors. The sensor for signal acquisition is the *MPU6050 module*, which contains, in a single, integrated chip, a 3-axis micro electro–mechanical system (MEMS) accelerometer and a 3-axis MEMS gyroscope; it provides a high sampling frequency as well as good precision thanks to a 16-bit analog-to-digital (AD) converter for each channel. The acceleration data generated by the MPU6050 module are acquired at a rate of 100 Hz, and the results are transmitted via USB to a laptop, where the subsequent time–frequency analysis, feature extraction and distress classification—as illustrated in Figure 1—take place using the Matlab environment.

During time–frequency analysis, acceleration along the vertical axis—after subtracting the mean—is processed over subsequent 12.5 s frames. This time duration was chosen to achieve a reasonable trade-off between low-latency and sufficiently long acquisition for proper pavement distress classification. In particular, the STFT of the current acceleration frame is performed by a windowing process, in which each window includes 50 samples (corresponding to 0.5 s) with 75% overlap between adjacent windows. The resulting magnitude of the STFT is then divided into 65 × 100 blocks (Figure 3b), and the energy of each submatrix is computed as in (Equation 1). By processing the energy of the STFT blocks, the coefficient of variation (CV) and the entropy (*H*) of the entire STFT matrix, as expressed in (Equation 7) and (Equation 8), are determined and, successively, fed to the classification module.

In order to expand the experimental assessment, the acceleration sensor was rigidly attached to the car’s body in two locations: the dashboard (Figure 4a) and the floorboard (Figure 4b).

Experimentation was carried out in urban and suburban areas in Florence (Italy) and involved two environmental scenarios. The first one was an urban scenario (see Figure 5a), where several itineraries were considered, including various types of road surface distresses and anomalies. The second one was a suburban scenario (Figure 5b), where most of the data were from better road-surface conditions. In our experiments, the car traveled at an approximately constant speed of 30 km/h; therefore, considering the acquisition rate and the duration of the analysis frame, one frame covers about 100 m and an acceleration sample was recorded every 8 cm. A total of 200 frames were collected, approximately equally distributed for each sensor location and for each class of distress under study, as described in Section 5.

## 7. Experimental Results

In this Section, the performance of the proposed approach is presented and discussed. In order to reveal the characteristics related to different road pavement anomalies in the time–frequency domain, some preliminary and qualitative considerations can be made by observing the STFT plots of the acceleration signal presented in Figure 6.

It can be noted that in the case of a road pavement surface in good condition (Figure 6a) the signal power is distributed quite uniformly over the time–frequency domain, with no power concentration at particular frequency values. The presence of potholes or manhole covers in the urban road section is instead clearly identifiable as a peak of power at a particular time and scattered over all frequencies, as shown in Figure 6b. Moreover, Figure 6c plots the case of a signal acquired while crossing fatigue cracking and rutting, in which the effect of road pavement distress is less intense than the previous case, but is more evenly distributed over a longer period of time.

The results obtained by cluster analysis of the proposed system are shown in Figure 7, which groups the elements belonging to each class in the feature (H,CV) space for both sensor locations under study (car’s dashboard and floorboard). As can be seen, the classes of short-time distress, long-time distress and no-distress can be quite clearly identified. By carefully observing the distribution of the experimental data, both features considered, *H* on the vertical axis and CV on the horizontal axis, have a high discrimination power.

As expected, data included in the short-time distress cluster represent road pavement conditions with high values of CV and low values of entropy (*H*) because the presence of acceleration peaks produces a high variation and low value of entropy of the signal.

The long-time distress cluster, compared to the previous one, is characterized by intermediate values of CV and *H* produced by fatigue cracking, an event that creates an acceleration signal that varies for longer time intervals with a lower intensity than that of potholes and manhole covers.

Lastly, the no-distress cluster shows the highest entropy values and the lowest CV values in the grouping plot. In this case, the accelerometer behavior is comparable to that of a noise source, and the absence of remarkable acceleration peaks leads, in fact, to low variation and high entropy of the signal.

In order to avoid overfitting and to properly evaluate the accuracy of the implemented classification algorithms, a 10-fold cross-validation scheme was applied. The advantage of this method is that all observations are used for both training and validation, and each observation is used for validation exactly once [39].

The performance results in terms of predictive accuracy on the complete data collection (car’s dashboard and car’s floorboard sensor locations) are presented in Figure 8a for each classification model and each model complexity.

The SVM algorithm achieved the best performance of the classifiers, with 86.4% accuracy granted by the linear kernel model.

In Figure 8b, the confusion matrix of the best-performing model is provided. On the confusion matrix plot, the rows correspond to the true class and the columns correspond to the predicted one. The diagonal cells correspond to observations that are correctly classified. The off-diagonal cells correspond to incorrectly classified observations. The rows at the bottom of the plot show the percentages of all the examples predicted to belong to each class that are correctly and incorrectly classified. These metrics are often called the *precision* (or *positive predictive value*) and *false discovery rate*, respectively.

It can be observed that the false discovery rate for long-time distress (23%) is significantly higher than for the other classes. This could be related to light potholes and fatigue cracking both being present at the same time in road sections with bad pavement condition, so misleading the linear SVM classifier to sporadically mix up long-time and short-time distress classes. Moreover, the results confirm the separation of clusters in the feature space (Figure 7), where the long-time distress class is adjacent to the others and consequently harder to accurately identify. On the other hand, the greater separation between the short-time distress and no-distress classes obtains excellent results in terms of precision and false discovery rate for the correct recognition of these road conditions. In fact, only 10% of short-time distress cases and 8% of no-distress ones are incorrectly assigned to the long-time distress class.

The impact of sensor position within the vehicle on performance can be evaluated by considering the classification results for the data collected on the car’s dashboard and floorboard separately. Figure 9 shows the distribution of the features in the feature space for data collected on the car’s dashboard and floorboard.

Comparing the results in Figure 10 and Figure 11, it can be seen that the floorboard location yields better classification rates for all classes of road pavement distress under study (97%, 84%, and 97% for short-time distress, long-time distress and no-distress, respectively), with the highest predictive accuracy value (91.9%) achieved by the cubic kernel model of the SVM algorithm. This difference in performance may be due to the fact that, as shown in Figure 2, the vertical component of the acceleration signal is less noisy when the sensor is fixed to the car’s floorboard, allowing the classifier to be more sensitive to different types of distress and better recognize, in particular, short-time distress. Nevertheless, the results obtained by the dashboard location, which corresponds to the typical positioning of mobile devices inside vehicles, are still very encouraging from the point of view of future application of the proposed techniques in a mobile crowdsensing scenario. In a context with a large number of data, in fact, great accuracy in recognizing the no-distress class (100%) could be crucial to reduce the adverse effects caused by false positives, i.e., when the classification results incorrectly indicate the presence of distress.

## 8. Conclusions

In this work, an effective and low-cost system for real-time monitoring of road pavement conditions based on data provided by on-car accelerometers has been proposed. A short-time Fourier analysis performed on the acceleration signal allowed the extraction of signal energy information from the time–frequency domain. Then, the coefficient of variation and the entropy of the signal energy were chosen as the features to distinguish three classes of road pavement conditions: short-time distresses such as potholes and manhole covers, long-time distresses such as fatigue cracking and rutting, and, finally, the absence of distresses. Three supervised machine learning classification families were considered: Decision Tree (DT), Support Vector Machine (SVM) and *k*-Nearest Neighbor (*k*NN); for each classification algorithm, three models were selected based on computational load and predictive accuracy. Measurement was performed both to collect real data and to evaluate the performance of the classification algorithms by comparing two on-car accelerometer locations (car’s dashboard and floorboard). The SVM classifier was the most accurate, in particular for the car’s floorboard sensor location, with 97%, 84% and 97% accuracy for short-time distress, long-time distress and no-distress, respectively.

## Figures and Tables

**Figure 1 sensors-22-03788-f001:**
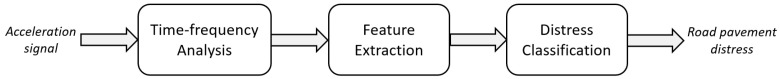
Block diagram of the proposed system.

**Figure 2 sensors-22-03788-f002:**
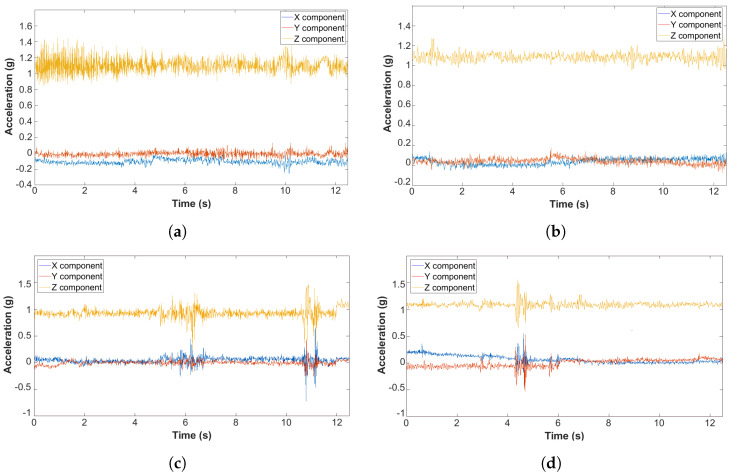
Examples of acceleration signals for different road pavement conditions and different on-car sensor placements. (**a**) Good pavement surface (car’s dashboard); (**b**) Good pavement surface (car’s floorboard); (**c**) Presence of potholes (car’s dashboard); (**d**) Presence of potholes (car’s floorboard).

**Figure 3 sensors-22-03788-f003:**
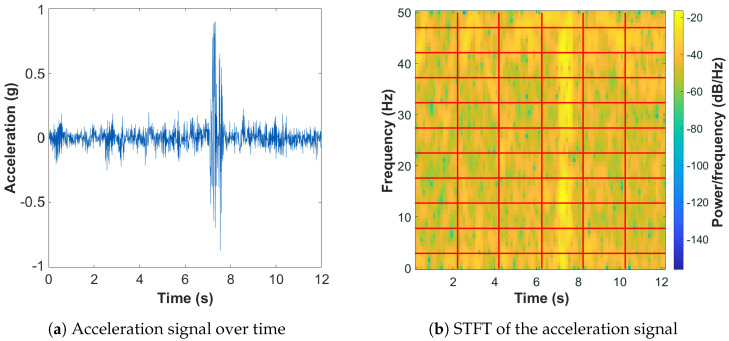
Example of high-pass filtered acceleration signal perpendicular to the road surface acquired during a period of 12 s (**a**) and of the magnitude of its STFT (**b**).

**Figure 4 sensors-22-03788-f004:**
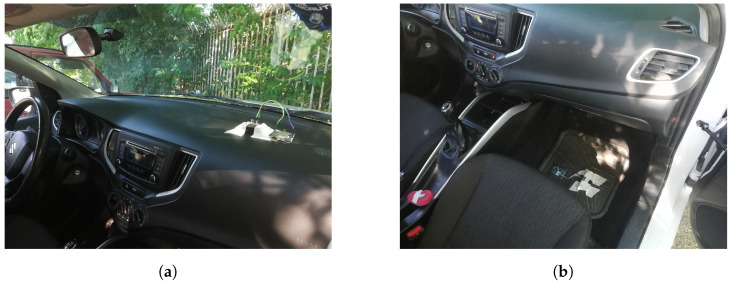
Onboard locations of the acceleration data acquisition system. (**a**) Acceleration sensor placed on the car’s dashboard; (**b**) Acceleration sensor placed on the car’s floorboard.

**Figure 5 sensors-22-03788-f005:**
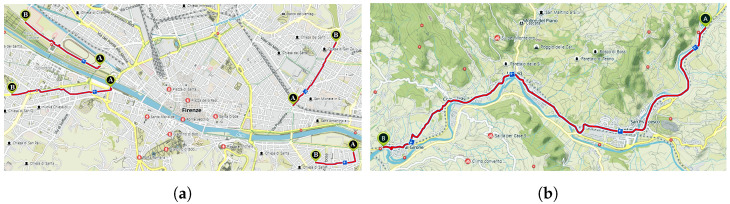
Maps of the acquisition scenarios in Florence. (**a**) Urban scenario; (**b**) Suburban scenario.

**Figure 6 sensors-22-03788-f006:**

Plots of the STFT of acceleration frames for different road pavement conditions. (**a**) Good pavement condition; (**b**) Pothole distress; (**c**) Fatigue cracking distress.

**Figure 7 sensors-22-03788-f007:**
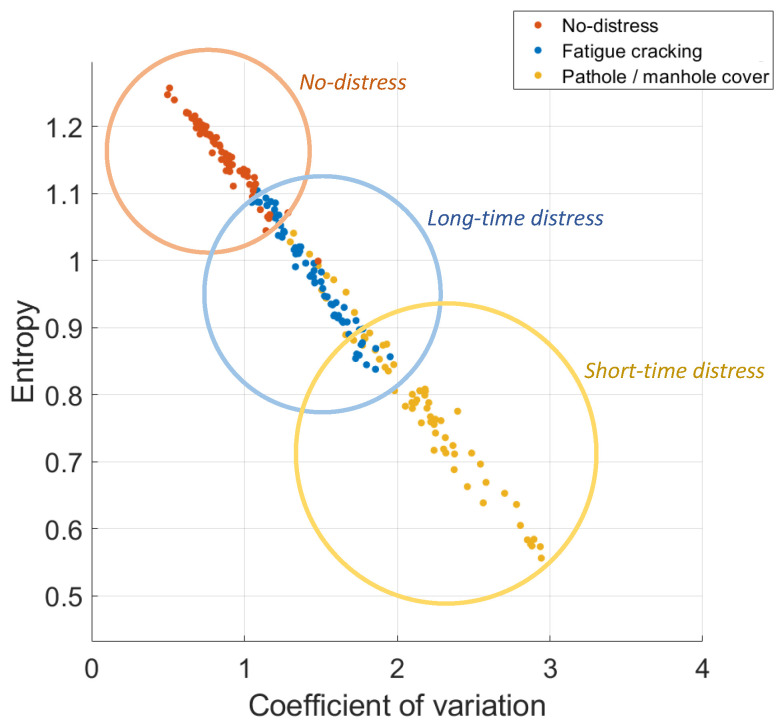
Distribution in the feature space of the data collected on both the car’s dashboard and the car’s floorboard.

**Figure 8 sensors-22-03788-f008:**
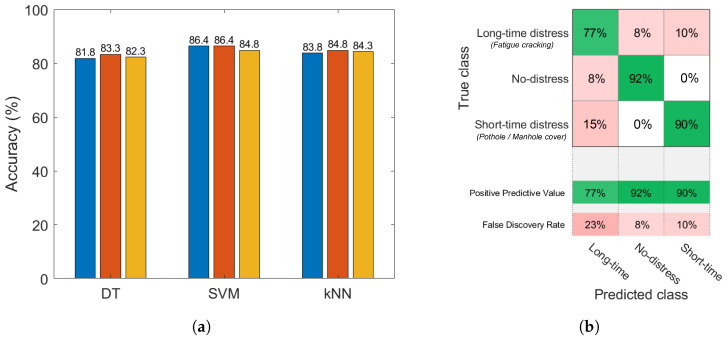
Classification results obtained by merging the data collected on both the car’s dashboard and the car’s floorboard; predictive accuracy of the considered supervised learning classification families: DT (2, 5, 10 splits), SVM (linear, quadratic, cubic kernels), *k*NN (1, 10, 30 neighbours) (**a**); confusion matrix of the linear SVM classifier (**b**).

**Figure 9 sensors-22-03788-f009:**
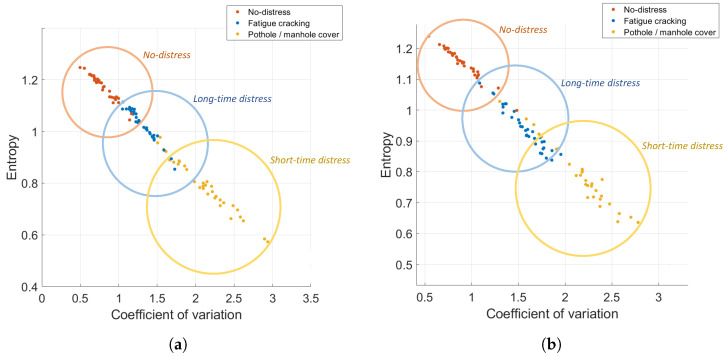
Distribution in the feature space of the data collected on the car’s dashboard (**a**) and floorboard (**b**).

**Figure 10 sensors-22-03788-f010:**
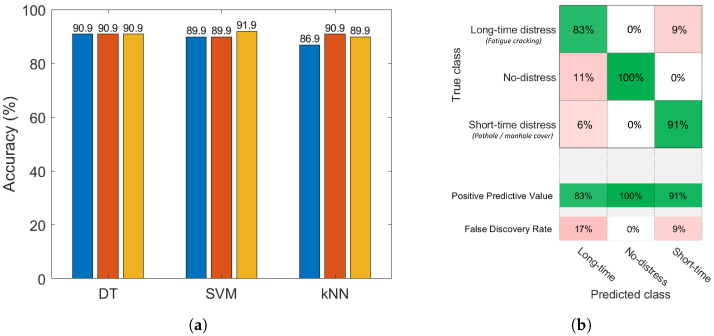
Classification results based on data collected from the car’s dashboard; predictive accuracy of the considered supervised learning classification families: DT (2, 5, 10 splits), SVM (linear, quadratic, cubic kernels), *k*NN (1, 10, 30 neighbors) (**a**); confusion matrix for the cubic SVM classifier (**b**).

**Figure 11 sensors-22-03788-f011:**
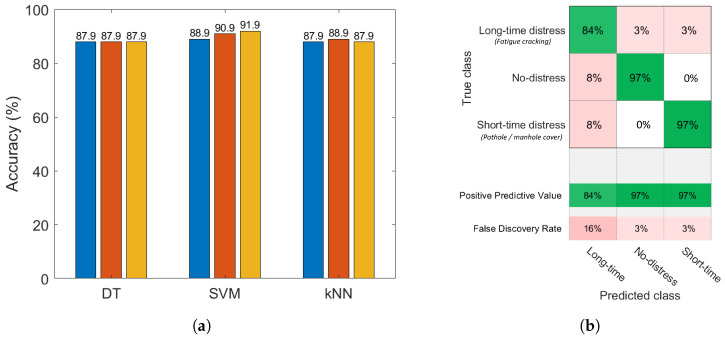
Classification results based on data collected from the car’s floorboard; predictive accuracy of the considered supervised learning classification families: DT (2, 5, 10 splits), SVM (linear, quadratic, cubic kernels), *k*NN (1, 10, 30 neighbours) (**a**); confusion matrix for the cubic SVM classifier (**b**).

## Data Availability

Not applicable.

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
