# Peer review of "Road Surface Anomaly Assessment Using Low-Cost Accelerometers: A Machine Learning Approach"

_sensors, 2022, doi:10.3390/s22103788_

Round 1

Reviewer 1 Report

Please find my comments in the attached file. 

Author Response

Manuscript ID: sensors-1688925: Reply to Reviewers

We would like to thank the Associate Editor and the Reviewers for their comments and remarks that helped us to improve the technical soundness and the clarity of this manuscript. In accomplishing the revision, all the comments have been taken into account and the content of the paper has been modified accordingly. All the changes to the manuscript have been highlighted in blue color. A point-by-point response is included below.

Responses to the comments of Reviewer1

General Comment  – Review 1

Overview and general recommendation: The paper presents a study on real-time monitoring of road conditions based on acceleration signals. The novelty of the work along with justifications regarding the adaptation of this procedure over existing contemporary methods should be coherent which is primarily missing from this manuscript. Before proceeding to the next stage, I would suggest the authors review the following queries in detail in an attempt to clarify the ambiguity in the draft. I strongly recommend the author to remodel the draft, address all the comments, and prepare a detailed revision with proper annotations.

Response  – We thank the Reviewer for the helpful comments. We have profoundly revised the manuscript according to the suggestions provided by the Reviewer and we hope to have fully answered all the remarks raised to the original manuscript.

Major Comments Review 1

Comment 1

Line 37: It is expected that the authors would provide a more global context on unsafe transportation beyond the Italian framework. There are several openly available websites that provide information on transportation faults and damages which could be provided as a short summary.

Response 1 – The authors would like to thank the Reviewer for this comment. As suggested, the introduction has been changed and now it includes a more detailed report about the effects of transportation faults and damages. In particular, the situation in Europe, USA and around the world in general is cited.

Comment 2 – Three paragraphs into the paper and the main objective of the work is not clearly pointed out.

It is suggested that the authors consider re-framing the sentences in order to provide a succinct information on what it is they are actually trying to achieve..

Response 2 – As suggested by the Reviewer, the introductory section has been re-organized and, now, in the revised manuscript, the objective of our study is explicitly stated in the first paragraphs of the Introduction.

Comment 3 – The novelty of the work along with the motivation for carrying out this research should be mentioned..

Response 3 – The motivation and the novel contributions of our study have been emphasized in a more incisive way in the introductory section.  More specifically, the novelty of the proposed methodology has been briefly discussed in relation also with the various vision/vibration-based methods and literature findings, whereas the motivations at the basis of this study are given with reference to the administrative and economic framework in which these techniques can find an application. Furthermore, the connections with recently proposed advanced methodologies, such as diffuse sensing (crowdsensing), are better highlighted.

Comment 4 – Line 54: The authors have missed out on the vibration-based crack detection techniques in roads that have been successfully implemented through eigen perturbation algorithms. The methods have proven to be industrial standards as real-time formulations that are computationally cheap and are able to identify hair-line cracks in roads of various surface roughness conditions. The authors should provide a narrative around this.

Response 4 – The authors would like to thank the reviewer for this suggestion. We have added a paragraph in the literature review addressing eigen-based methods and the bibliography has been integrated accordingly.

Comment 5 – For citizen science approaches, how can the methods be used if smartphones were carried by users inside cars for a crowdsourced ensemble of data?

Response 5 – The authors would like to thank the reviewer for this suggestion. In our study, the acquisition apparatus is based on embedded systems connected to sensing devices: this can be viewed as a development platform in order to test the mathematical tools, methods and algorithms as well as to evaluate their performance. The employment of smartphones, using their own accelerometer and running the proposed methods in suitable apps, may be seen as an evolution of the proposed system in a framework of mobile crowdsensing. Thus, the present study may be seen as the initial step of a more complex crowdsourcing system able to exploit the data from a multitude of sensors within the road network (e.g., cars, light vehicles, etc.). A brief discussion about this fact has been added into the introduction.

Comment 6 – Line 64: Please explain the rationale behind adopting STFT as the preferred candidate for analysing acceleration signals

Response 6 – The signals taken into account in this study are definitely nonstationary. For this reason, tools like the short-time Fourier transform (STFT), wavelets, etc., are perfect candidates for their analysis. The STFT is, maybe, the simplest tool, but, nevertheless, it has been found to be very effective in extracting the necessary features, both in the time and the frequency domain, able to discriminate the classes of interest in our work.

Comment 7 – Line 202: The assumption that accelerometers are installed on the car body is vague and misleading.

The precise locations of the accelerometer should be previously known in order to carry out

field tests

Response 7 – We thank the Reviewer for remarking on this fact. In this study, an accelerometer is positioned within the car, on either the dashboard or the floorboard, as described in  Section 6, where photographs of the positioning of the acquisition systems are shown. In our opinion, these pictures should fully clarify this point.

Comment 8 – Please explain the quarter car model in detail if the same has been adopted in the study

Response 8 –  The quarter car model is cited as a tool to explain how vibrations are transmitted from the road surface to the body and the interiors of the car, but, actually, it was not used for our derivations and experiments.

Comment 9 – Lines 270-275: Please provide sufficient evidence for each of these assumptions in the time and the frequency domain

Response 9 – The assumptions expressed in this paragraph are basically drawn from the experience and from an intuitive analysis of our data. It is well-known that impulse-like and periodic signals produce broadband and narrowband Fourier transforms, respectively. Due to the isolated spikes induced by potholes onto the acceleration signal, this class is associated with the former type (impulse-like) of signals. On the other hand, extensive pavement cracking is expected to be characterized by a cyclic roughness in the road surface and, thus, it is associated with the latter type (periodic-like) of signals. The measures of sparsity proposed to discriminate the different classes were based on these considerations. Even though the above discussion was derived only on an empirical basis, they were confirmed by experiments in a supervised learning context. In the revised manuscript, the lines of the text have been slightly rephrased.

Comment 10 – Line 302: On what basis are these ML-classifiers chosen? the motivation behind adopting the approaches is unclear.

Response 10 –  The classifiers were chosen based on observing the structure of data in the feature space. Even though Decision Tree (DT), Support Vector Machine (SVM) and k-Nearest Neighbour (kNN) are among the simplest ML tools for supervised learning, experimental tests confirmed that the classification rates achievable by using such classifiers are definitely high and we deem that they can be considered as satisfactory for the purposes of this study.

Comment 11 – Figure 8: If the data would not have been represented with markers of different colors, how would the change in variation be certain?

Response 11 – Figure 8 represents, with different markers, the classes in the feature space. This type of representations is commonly used in supervised classification and its scope is trying to roughly identify structures in the data able to suggest classifications strategies.

Comment 12 – Figure 9b: Please explain the diagonal entries of the confusion matrix and their corresponding physical implications in the real-world

Response 12 – The diagonal entries of the confusion matrix represent the positively predicted values of each classifier, that is its ability to recognize a sample of data as belonging to a class; in other words, they are related to the accuracy of the classifier. In the revised version of the manuscript, some details about confusion matrix implications have been added.

Comment 13 – Cost considerations for the proposed approach have not been clearly demonstrated. The claims of a ’low-cost’ system is not thoroughly evident in the manuscript and needs to be discussed in detail. A steady comparison with a real-time counterpart – such as eigen perturbation theory – can be carried out to provide evidence of a low-cost regime that has been potentially developed through this study

Response 13 – The  term “low-cost” associated with the proposed system is related both to hardware equipment and implementation complexity of algorithms. As to HW requirements, the use of Arduino platforms permits the system to be realized with a very limited cost. As to the SW requirements, the use of the STFT is also competitive thanks to the availability of fast implementations.

Minor Comments Review 1

Comment 1 – Review 1

In the context of this manuscript, how is ’real-time’ defined?

Response 1 – A system works in “real-time” if the processing time of a data segment is shorter than the data segment duration. Even though in our tests signals were sampled and stored on a computer in order to process them offline (to test and evaluate the different algorithms), the computational burden is such that the system, once the classification method has been identified, can work in real-time.

Comment 2 – Figure 1: Please obtain necessary permissions for reproducing these figures, in case the authors are not directly involved in their acquisition.

Response 2 – The authors would like to thank the Reviewer for this comment. In the revised manuscript, we have eliminated Figure 1 and we have updated the bibliography reagarding manual road distress classification.

Comment 3 – Figure 3, 4: Please use bolder and larger fonts on the axes for each of these illustrations.

Response 3 – As suggested, the fonts have been changed and now, in the revised manuscript, all the referenced Figures present uniform font dimensions.

Comment 4 – It is a general practice to use bold and uppercase for matrices, italics and uppercase for vectors, and lowercase for scalars. The authors can choose to follow this notation.

Response 4 – In this study, matrices are not directly involved in manipulations, but, rather, only their (scalar) entries are considered for the subsequent classification; in other words, the quantities defined in the manuscript are indeed matrix elements and the features derived from them. Even though our notation is simple, we deem that it is sufficiently clear for the purposes of this study.

Reviewer 2 Report

This manuscript reports an interesting low-cost system for a real-time monitoring of the road pavement conditions considering the data obtained through on-car accelerometers. The data were processed using short-time Fourier analysis to extract the signal energy information in the time-frequency domain. Authors used the coefficient of variation and the entropy of the signal energy to measure the difference between three types of road pavement conditions. The proposed system could be used to detect different types of anomalies on the road surface.

1.-Which are the main limitations of the proposed system to detect anomalies on the road surface?

2.-Which are the main advantages of the proposed system in comparison with other system reported in the literature?

3.-Which are the separation distances of the two positions of the on-car accelerometers with respect to the road surface?

4.-Authors should include more technical information of the car type and its damping system. A bad damping system of the car could affect the measurements of the accelerometers.

5.-Authors should include more discussion of the behavior of the experimental results for the three types of anomalies.

6.-What is the future research work?

Author Response

Manuscript ID: sensors-1688925: Reply to Reviewers

We would like to thank the Associate Editor and the Reviewers for their comments and remarks that helped us to improve the technical soundness and the clarity of this manuscript. In accomplishing the revision, all the comments have been taken into account and the content of the paper has been modified accordingly. All the changes to the manuscript have been highlighted in blue color. A point-by-point response is included below.

Responses to the comments of Reviewer2

General Comment  – Review 2

This manuscript reports an interesting low-cost system for a real-time monitoring of the road pavement conditions considering the data obtained through on-car accelerometers. The data were processed using short-time Fourier analysis to extract the signal energy information in the time-frequency domain. Authors used the coefficient of variation and the entropy of the signal energy to measure the difference between three types of road pavement conditions. The proposed system could be used to detect different types of anomalies on the road surface.

Response – We thank the Reviewer for the helpful comments. The Reviewer has fully achieved the objective of our work.

Major Comments Review 2

Comment 1

Which are the main limitations of the proposed system to detect anomalies on the road surface?.

Response 1 – A main limitation of the proposed method may be, as with every vibration-based approach, the inability to measure road-surface damages in areas other than the vehicle wheel paths,  with the consequence of not being able to identify the size of the road-surface damage across the entire road section. This issue is mitigated by the crowdsensing paradigm. A brief discussion about this has been added in the Introduction.

Comment 2 – Which are the main advantages of the proposed system in comparison with other system reported in the literature?

Response 2 – The proposed system represents an answer to the problem of the high-cost monitoring process based on video/image processing. Moreover, the proposed method aims at not only detecting road surface anomalies but also to classify the different types of degradations. A brief discussion about this has been added in the Introduction.

Comment 3 – Which are the separation distances of the two positions of the on-car accelerometers with respect to the road surface?

Response 3 – The floorboard and dashboard positions are about 40 cm and 120 cm distant from the road surface.

Comment 4 – Authors should include more technical information of the car type and its damping system. A bad damping system of the car could affect the measurements of the accelerometers.

Response 4 – The car used for the experimental tests was a Suzuki Baleno with MacPherson (anterior wheels) and torsion bar (posterior wheels) suspension systems. The vehicle was in perfect maintenance condition, so that it is assumed that suspensions were fully efficient.

Comment 5 – Authors should include more discussion of the behavior of the experimental results for the three types of anomalies

Response 5 – The authors would like to thank the Reviewer for this comment. As suggested, the experimental results section has been changed and now it includes a more detailed discussion about the performance results for the three types of road anomalies.

Comment 6 – What is the future research work?

Response 6 – The employment of smartphones, using their own accelerometer and running the proposed methods in suitable apps, may be seen as an evolution of the proposed system in the framework of mobile crowdsensing. Thus, the present study may be seen as the initial step of a more complex crowdsourcing system able to exploit the data from a multitude of sensors within the road network (e.g., cars, light vehicles, etc.). A brief discussion about this fact has been added into the introduction.

Round 2

Reviewer 1 Report

Besides a small comment, I am happy with the responses to review queries. The authors can consider removing reference #10 as reference #9 already justifies the technical background of the corresponding content. 

Reviewer 2 Report

Authors have improved their manuscript considering the comments of reviewer.